# Assessment of established techniques to determine developmental and malignant potential of human pluripotent stem cells

The International Stem Cell Initiative[#]

The International Stem Cell Initiative compared several commonly used approaches to assess human pluripotent stem cells (PSC). PluriTest predicts pluripotency through bioinformatic analysis of the transcriptomes of undifferentiated cells, whereas, embryoid body (EB) formation in vitro and teratoma formation in vivo provide direct tests of differentiation. Here we report that EB assays, analyzed after differentiation under neutral conditions and under conditions promoting differentiation to ectoderm, mesoderm, or endoderm lineages, are sufficient to assess the differentiation potential of PSCs. However, teratoma analysis by histologic examination and by TeratoScore, which estimates differential gene expression in each tumor, not only measures differentiation but also allows insight into a PSC's malignant potential. Each of the assays can be used to predict pluripotent differentiation potential but, at this stage of assay development, only the teratoma assay provides an assessment of pluripotency and malignant potential, which are both relevant to the pre-clinical safety assessment of PSCs.

#A full list of consortium members appears at the end of the paper.

The capacity to differentiate into derivatives of all three embryonic germ layers are the central defining feature of all pluripotent stem cells (PSC), but assessing this property remains a challenge for human cell lines. PSC were first recognized as embryonal carcinoma (EC) cells in teratocarcinomas, germ cell tumors that also contain a wide array of somatic tissues[1–4]. In a classic experiment, using a teratocarcinoma of the laboratory mouse characterized by Stevens[5] Kleinsmith and Pierce[6] provided the first functional demonstration of pluripotency by showing that single cells from ascites-grown embryoid bodies (EBs) could generate tumors containing EC cells together with somatic tissues. The connection between teratocarcinoma and normal embryos was subsequently established by experiments showing that embryos transplanted to extra-uterine sites inevitably develop into teratomas or retransplantable teratocarcinomas[7,8]. The discovery that murine EC cells can participate in embryonic development when transferred to early mouse embryos to give rise to chimeric mice[9] led to the realization that EC cells have the developmental capacity of cells of the inner cell mass. This laid the groundwork for the derivation of embryonic stem (ES) cells from mouse embryos[10,11] and later from human embryos[12] and of induced PSC (iPSC) from differentiated human cells[13,14].

In assessing mouse ES or iPS cell lines, pluripotency is functionally defined from the PSC. However, for human PSC, be they ES or induced pluripotent stem cells (iPSC) cells[13,14], this fundamental assay is by the cell line's ability, when transferred to a preimplantation embryo, to form to a chimeric animal in which all of the somatic tissues and the germ line include participating cells not available. Moreover, a variety of well characterized PSC, from both mice and primates have only a limited ability to participate in chimera formation, even though they can differentiate into tissues of all three germ layers in teratoma and in vitro assays[15]. With the advent of technologies for producing large numbers of human PSC[16,17], some destined for clinical applications, the need for rapid and convenient assays of a specific PSC's pluripotency and differentiation competence has become paramount.

The purpose of this study was to provide an authoritative assessment of several established alternative techniques for determining the developmental potential of human PSC lines. The PluriTest® assay[18] (www.pluritest.org), is a bioinformatics assay in which the transcriptome of a test cell line is compared to the transcriptome of a large number of cell lines known to be pluripotent. This test can be carried out rapidly with small numbers of cells, an important consideration in the early stages of establishing new PSC lines. PluriTest is able to exclude cells that differ substantially from undifferentiated stem cells, but does not directly assess differentiation capacity. Complementing PluriTest's focus on the undifferentiated state, various methods have been developed to monitor differentiation of the PSCs themselves in vitro, including protocols that induce spontaneous differentiation of cells in either monolayer or suspension culture, or directed differentiation under the influence of specific growth factors and culture conditions that promote the emergence of particular lineages[19,20]. One of the most common approaches has been the use of differentiation in suspension culture, when clusters of cells undergo differentiation to form embryoid bodies (EB), often with some internal structure apparent[21]. EB differentiation has also been combined with gene expression profiling and bioinformatic quantification of gene signatures, giving rise to the pluripotency scorecard assay[22]. Further development of this scorecard defined a panel of 96 genes that identified the differentiation capacity of a given cell line more quantitatively than the typical histology-based teratoma assay[23]. The teratoma assay has long been regarded as the 'gold standard' for assessing human

PSC pluripotency. Not only do truly pluripotent cells generate a very wide array of derivatives in these tumors, but they are also often organized into organoid structures reminiscent of those that appear during embryonic development[24]. However, both the production of teratomas as xenografts, and their detailed analysis, which requires appropriately trained specialists, is costly and time consuming, and may be limited by concerns over animal welfare. Moreover, the teratoma assay, as routinely performed, does not yield quantitative information on lineage differentiation potential[25], although gene expression analysis of the teratomas themselves can supply more definitive analysis.

In the current International Stem Cell Initiative (ISCI) study, following discussion at an ISCI workshop attended by about 100 members of the human PSC research community, we carried out a comparison of these approaches for assessing pluripotency by conducting a series of assays with human PSC lines, both ES and iPS cells. PluriTest was used to assess the transcriptome of the undifferentiated cell lines. For the EB assay, we chose one widely used approach, the 'Spin EB' system[21] and used an adapted lineage scorecard methodology[22] to assess the results. The Spin EB method provides for control of input cell number and good cell survival, and allows for differentiation under neutral conditions and under well-defined conditions expected to promote differentiation towards ectoderm, mesoderm or endoderm. Differentiation in teratomas was appraised by both histological examination and by "TeratoScore", a computational quantitation of gene expression data derived from teratoma tissue[26].

These blinded analyses, conducted by independent experts on PSC-derived samples in four highly experienced laboratories, shows that each of these methods can be used to indicate pluripotency and that each is able to detect some variation in developmental potential among the cell lines. The choice of which method(s) should be used must be dictated by the biological question posed and the future use of the PSCs in question. We propose a schema outlining the choice of methodology for particular applications.

## Results

**Experimental design**. To compare PluriTest, EB differentiation and teratoma, assays under conditions that would reflect variability between laboratories and cell lines, four separate, expert laboratories in four countries carried out these studies on each of three different, independent PSC lines and a fourth cell line, H9 (WA09)[12], which was common to all (Supplementary Table 1). All the experimental material was processed centrally, with high-throughput RNA sequencing (RNA-seq), quantitative real-time PCR and histology, as well as bioinformatics analyses carried out by single-specialized laboratories. In total, we compared results from 13 PSC lines (seven ESC and six iPSC lines).

**Genetic integrity**. It has been suggested that karyotypically variant PSC might be associated with persistence of undifferentiated cells in xenograft tumors[27,28]. As an important adjunct to the differentiation studies we took several approaches to assess the genetic integrity of the cell lines. Prior to initiating the experiments, the four test laboratories confirmed that the cell lines they planned to use had normal diploid karyotypes, excepting NIBSC5, which carried a gain of the chromosome 20q amplicon that has been previously described[29]. Gene expression data also permitted evaluation of the genetic integrity of the cell lines at the time they were used in the experiments. Over- or under-representation of specific regions of the genome in the undifferentiated PSC lines was evaluated using expression karyotyping (e-Karyotyping)[30]. Of the 13 cell lines, only one, the ES cell line MEL1 $INS^{GFP/w}$, showed an aberrant e-karyotype containing

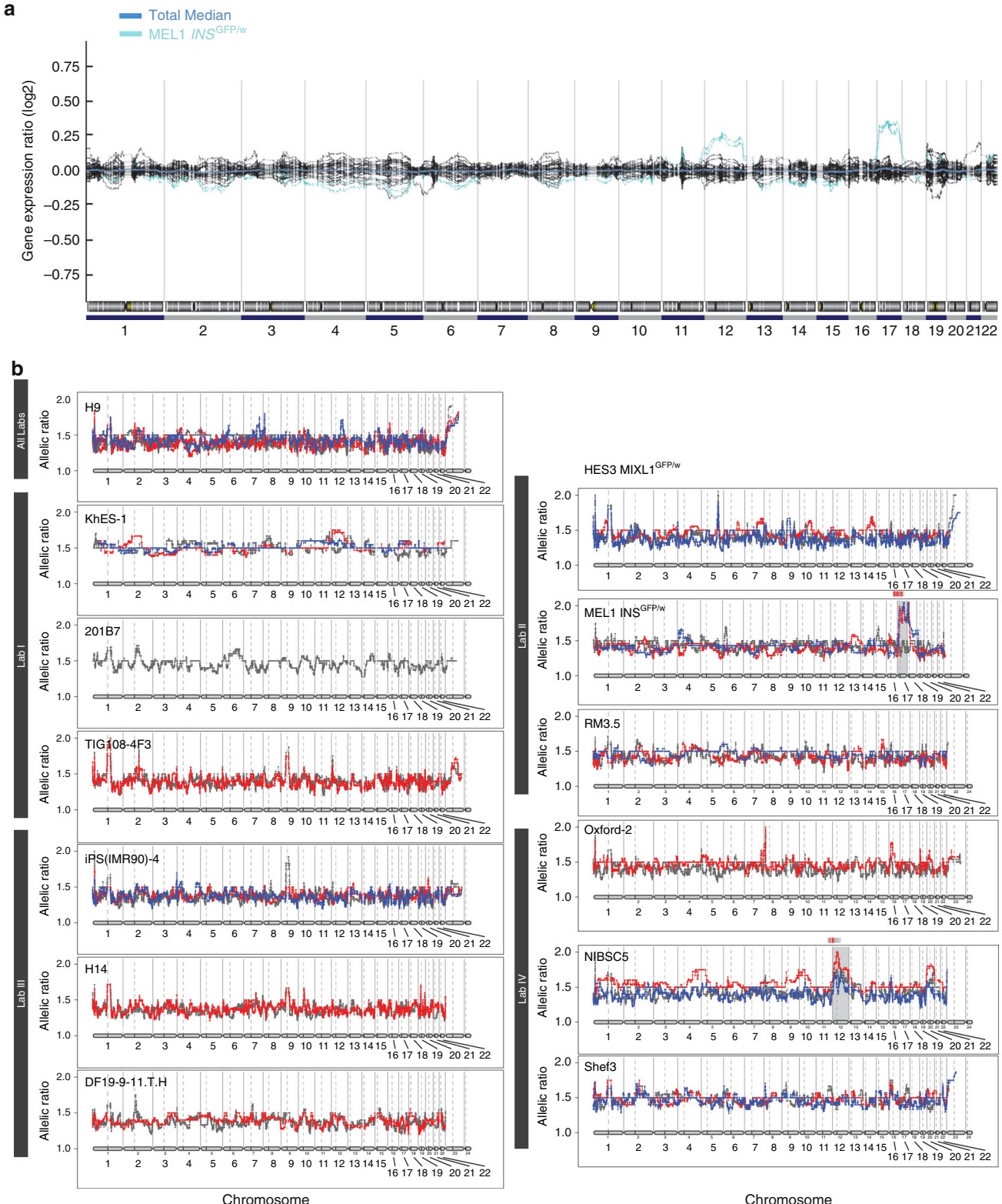

**Fig. 1** Detection of chromosomal aberrations in PSC and tumors using e-Karyotyping and eSNP-karyotyping. **a** e-Karyotyping: each line depicts the moving average plots of global gene expression in 13 different cell lines over 300-gene bins. The gene expression of 12 cell lines (black lines) was close to the total mean, suggesting a normal karyotype. In contrast, all replicates of the MEL1 *INS*^GFP/w (cyan) cell line showed considerable upregulation of genes from both chromosomes 12 and chromosome 17, suggesting that it harbors an additional copy of these chromosomes. **b** eSNP-karyotyping: detection of chromosomal aberrations in tumors using eSNP-karyotyping. Each line depicts the moving average (over 151 SNPs) of gene expression generated from RNA-seq data of tumor derived from 13 different cell lines (one plot per source cell line). Colors represent tumor replicates. Only tumors derived from MEL1 *INS*^GFP/w and NIBSC5 show an altered allele ratio in both replicates, suggesting an aberrant karyotype with additional copies of chromosomes 17 and 12, respectively

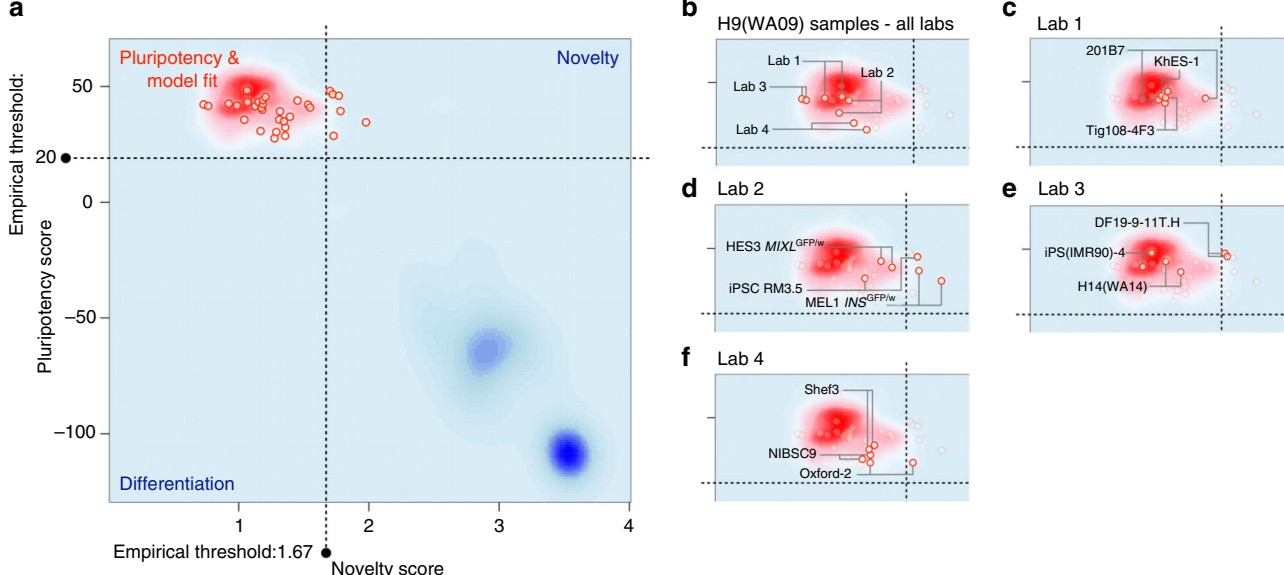

**Fig. 2** Pluritest. **a** All PluriTest results from this study (red circles) are based on normalization to the H9 samples and were plotted on the background of the empirical density distribution of all pluripotent (red cloud) and differentiated samples (blue clouds) in the PluriTest training dataset[18]. **b-f** highlight the subsets of samples included in this study: All results from the same hPSC line (H9) cultured at each laboratory (**b**). Samples from Lab 1 (**c**), Lab 2 (**d**), Lab 3 (**e**), Lab 4 (**f**) are highlighted specifically. All cell lines are above the Pluripotency Score threshold ($\theta_P >= 20$). Both replicates of two cell lines MEL1 $INS^{GFP/w}$ in **d** and DF19-9-11T.H in **e** score above the Novelty threshold ($\theta_N >= 1.67$) and thus would be highlighted for further investigation. Three cell lines show larger differences between the novelty scores of their respective replicate samples 201B7 in **c**, RM3.5 C in **d**, and Oxford-2 in **f**

extra copies of chromosomes 12 and 17 (Fig. 1a). These discrepancies from the test laboratories reports for NIBSC5 and MEL1 $INS^{GFP/w}$ most likely reflect the sensitivities of different assays for detecting low level genetic mosaicism[31] and the propensity of variants to overgrow the culture rapidly once they appear[32]. Consistent with this interpretation, the MEL1 $INS^{GFP/w}$ is known to exhibit karyotypic instability in culture (RM, EGS, and AGE, unpublished results). Because of the heterogeneous cell composition of teratomas a different methodology is required to evaluate the chromosomal integrity of the cells comprising them. eSNP-karyotyping enables a direct analysis of chromosomal aberrations by calculating the expression ratio of SNPs, making it less sensitive to global gene expression changes between different samples[33]. eSNP-Karyotyping of the teratomas indicated that most remained karyotypically diploid, but also revealed that teratomas derived from NIBSC5 had additional copies of chromosomes 12 (and perhaps 20), and that teratomas derived from MEL1 $INS^{GFP/w}$ carried an additional copy of chromosome 17, but not chromosome 12 (Fig. 1b). Extra copies of human chromosomes 12, 17, and 20 are recurrent changes in cultured PSCs, and have also been reported in human germ cell tumors[29]. These changes likely reflect a selective advantage conferred by extra copies of genes on these chromosomes to cells grown either in vitro or in vivo[34,35]. Taken together our results suggest that cultures of NIBSC5 and MEL1 INSGFP/w, but of none of the other 11 lines, were initially mosaic containing low levels of variant cells.

**PluriTest analysis.** PluriTest was used to assess the molecular similarity of the different undifferentiated cell lines to that of other known PSC lines. RNA samples were analyzed using the Illumina Human HT-12 v4 Expression BeadChip and subjected to the PluriTest algorithm[18]. PluriTest generates two summary scores from global gene expression profiles: a pluripotency score that predicts whether a cell sample is pluripotent based on the similarity of its gene expression signature to gene expression profiles of a large collection of human PSC; and a novelty score

that detects the presence of gene expression patterns usually not associated with human PSC. A pluripotent cell line is characterized as passing the PluriTest if it simultaneously exhibits a high Pluripotency and a low-novelty score. If the scores of a test cell line deviate from the empirically determined Pluripotency and Novelty thresholds, the sample is flagged for further investigation. As the original PluriTest algorithm was developed for an older Illumina BeadChip platform, it was adapted to a new platform using the H9 samples from all four laboratories as a control for technical variation (Supplementary Fig. 1). Analyzing samples with the updated PluriTest script, showed that at least one replicate of most lines assayed passed both PluriTest criteria (Fig. 2; Supplementary Fig. 1).

In the case of cell lines RM3.5 and Oxford-2, while we observed high-Pluripotency Scores in both replicates (Fig. 2), there was a large difference in the Novelty Scores between the two replicates, placing one replicate above the empirical threshold for the Novelty Score (1.67). A similar result was obtained for one of the two replicates from the 201B7 cell line. The differences in Novelty score observed between replicates could be due to technical failures of the array hybridization, or it could reflect differing extents of spontaneous differentiation in the cell line samples analyzed. Nevertheless, we concluded that all cell lines with one replicate below the empirical Novelty Score threshold passed PluriTest and are predicted to have pluripotent differentiation potential in vitro and in vivo. However, the PSC lines DF19-9-11T.H and MEL1 $INS^{GFP/w}$ did not pass the empirically determined Novelty Score threshold of 1.67, thus flagging them for further investigation. Interestingly, the MEL1 $INS^{GFP/w}$ PSC line did have an abnormal e-Karyotype (Fig. 1a, b), providing a possible explanation for its borderline results in PluriTest.

**Scorecard analysis of embryoid body differentiation in vitro.** The participating laboratories also subjected their cell lines to a standardized embryoid body (EB)-differentiation protocol under four different conditions: neutral, without the addition of exogenous growth factors that favored any particular lineage, and

directed conditions designed to promote initial differentiation into ecto-, meso-, or endoderm lineages, respectively[21]. It was anticipated that these protocols would be sufficient to direct differentiation toward the germ layer of interest but would not

necessarily support the generation of more mature cell types. Lysates from the resulting EBs were examined by qRT-PCR at 0, 4, 10, and 16 days of differentiation for expression of 190 genes (Supplementary data 2, 3) modified from the set used by

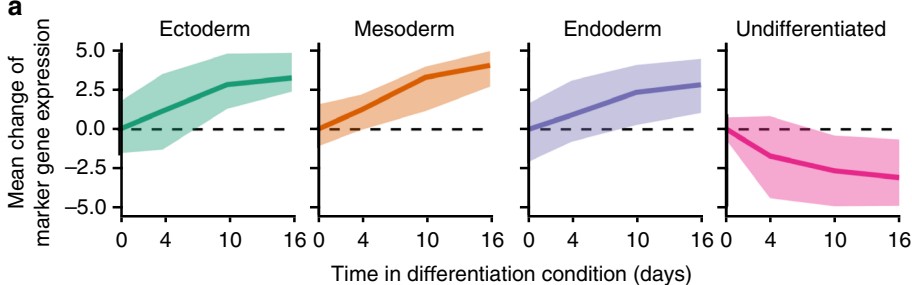

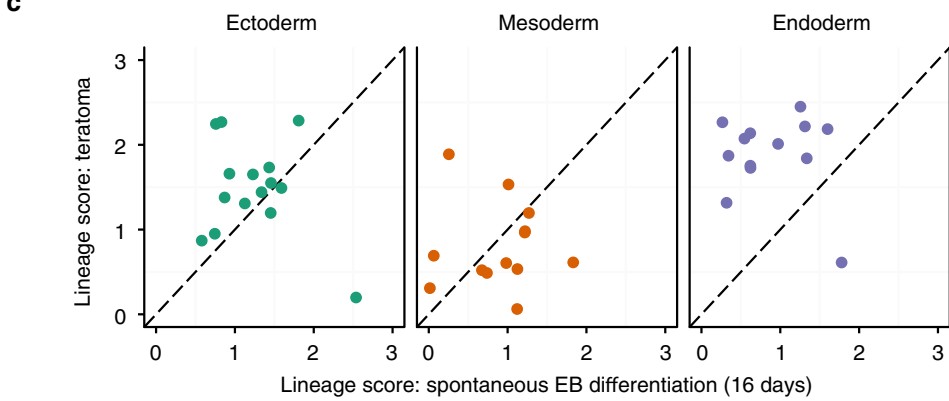

**Fig. 3** Differentiation potential and propensity in EBs. **a** The line plots show the mean $\log_2$ expression change (relative to day 0) of marker genes (Supplementary table 3) as a function of time and averaged over all cell lines. The expression change is shown under ectoderm conditions for ectoderm markers, mesoderm conditions for mesoderm markers, endoderm conditions for endoderm markers, and across all conditions for markers of undifferentiated cells. Shaded contours indicate the minimum/maximum observed value. **b** A summary table of the lineage scorecard evaluation of the "propensity" (spontaneous differentiation, left) and "potential" (directed differentiation, right) for each cell line (rows) to differentiate into the respective lineage (columns). Colors and symbols indicate increased (blue) and limited (grading of lighter blues) preference for expression of lineage- specific marker genes. +++: score >3; ++: score 2–3; +: score 1–2; +/–: score <1. nd not analyzed due to RNA failing quality control criteria. **c** Scatterplots contrasting the lineage score after 16 days of EB differentiation ("propensity"; x-axis) with the lineage score for teratomas derived from the same cell lines (y-axis). The lineage scores for ectoderm (left), mesoderm (center), and endoderm (right) marker expression are shown separately

**Table 1 Histology and teratoscore comparison of xenograft tumors**

| Lab | Cell Line | Cell Type | Xenograft Tumors | | | | | | | | | |
|---|---|---|---|---|---|---|---|---|---|---|---|---|
| | | | Histology[a] | | | | | RNA-seq | | | | |
| | | | | | | | | Teratoscore[b] | | | | |
| | | | Ecto | Meso | Endo | ECL | YS | Ecto | Meso | Endo | Extra Emb | ECL /YS[c] |
| Lab 1 | H9 | ES | +++ | + | + | – | – | nd | nd | nd | nd | nd |
| | KhES-1 | ES | +++ | + | + | + | + | ++ | + | + | + | + |
| | 201B7 | iPS | +++ | ++ | + | + | + | ++ | + | + | ++ | – |
| | Tig108 4f3 | iPS | ++ | ++ | ++ | + | + | ++ | ++ | + | ++ | + |
| Lab 2 | H9 | ES | ++ | ++ | + | – | – | ++ | ++ | +/++ | + | – |
| | HES3 MIXL1[GFP/w] | ES | +++ | ++ | + | – | – | ++ | ++ | + | + | – |
| | MEL1 INS[GFP/w] | ES | + | +++ | ++ | – | – | ++ | ++ | ++ | ++ | – |
| | RM3.5C | iPS | ++ | +++ | + | – | – | ++ | ++ | ++ | ++ | + |
| Lab 3 | H9 | ES | + | +++ | + | – | – | ++ | ++ | ++ | ++ | – |
| | H14 | ES | + | +++ | + | – | + | ++ | ++ | ++ | ++ | + |
| | DF19-9-11T.H | iPS | ++++ | + | + | – | + | ++ | +/++ | +/++ | +/++ | + |
| | iPS(IMR90)-4 | iPS | +++ | ++ | + | – | – | ++ | ++ | + | + | + |
| Lab 4 | H9 | ES | ++ | +++ | + | – | – | ++ | ++ | ++ | + | – |
| | Shef3 | ES | + | +++ | + | – | + | ++ | ++ | + | + | + |
| | Oxford-2 | ES | ++ | ++ | + | – | – | ++ | ++ | ++ | ++ | – |
| | NIBSC 5 | iPS | ++ | ++ | + | – | – | +/++ | ++ | ++ | +/++ | – |

[a] The presence of tissues scored as ectoderm (ecto), mesoderm (meso), endoderm (endo) in the histological examination of the tumors is summarized as median scores corresponding to the presence of the respective germ layers: '+' (0–25%), '++' (25–50%), '+++' (50–75%), and '++++' (>75%)
[b] For Teratoscore, the percentage of ectoderm, mesoderm, endoderm, and extraembryonic specific-gene expression is summarized in comparison to the mean percentage of 4 pilot, karyotypically normal teratomas: '+' (the pilot expression mean) '++' (similar to the pilot expression mean)
[c] The presence of undifferentiated cells (ECL) and/or yolk sac elements (YS), assessed by both histology and by RNA-seq analysis is indicated by '+', in cells that are highlighted in yellow

Bock et al.[22], to include genes characteristically expressed in undifferentiated PSC, extraembryonic endoderm, trophectoderm, early definitive ectoderm, mesoderm, and endoderm. For each lineage and for undifferentiated cells, we picked an equal number ($n = 15$) of marker genes for further analysis (Supplementary Table 2), by focusing on those genes with the strongest lineage-specific upregulation of genes in our dataset (Methods section). These marker genes were generally more highly expressed in EBs cultured under the corresponding differentiation conditions, while expression of markers of undifferentiated cells gradually dropped (Fig. 3a, Supplementary Fig. 2a). Gene expression was least variable 4 days after induction of differentiation compared to other time points (Supplementary Fig. 2b, c).

The lineage scorecard analysis was carried out as described previously[22] but with the refined gene set (Supplementary Table 3) and with one conceptual extension: the "potential" of cells to undergo differentiation into the three primary lineages under directed differentiation conditions was distinguished from their "propensity" to differentiate under neutral conditions. The "potential" of a cell to differentiate into a certain lineage was defined as the lineage score at 16 days of directed differentiation culture conditions. That is, ectoderm induction was used for ectoderm marker profiling, mesoderm induction for mesoderm markers, and endoderm induction for endoderm markers. The "propensity" (or inherent bias) of a cell line to undergo differentiation was calculated from the lineage scores (Methods section) of all marker sets after 16 days in neutral differentiation conditions.

Scorecard analysis resulted in three key observations (Fig. 3b, Supplementary Fig. 2a, b). First, in neutral culture conditions all cell lines had the propensity to upregulate ectoderm markers, but all cell lines also initiated mesoderm and endoderm expression

programs, though some (KhES-1, 201B7, RM3.5C, and H9 from Labs 2 and 4) had reduced propensities to form one or both of these latter lineages, an apparent bias not recapitulated in the teratoma assay (Table 1 below). Second, ectoderm-inducing and mesoderm-inducing conditions elicited strong, homogeneous expression signatures consistent with the expected directed lineage, while endoderm-inducing conditions elicited more variable responses, depending on both the cell line and on the laboratory, a result most marked in the Oxford-2 line. Third, the data suggest that, overall, all cell lines were capable of differentiating into representatives of all three lineages, although there were differences in how well and how consistently the PSC lines responded to these specific differentiation cues.

**Differentiation in xenograft teratomas in vivo.** Each laboratory produced between one and three xenograft tumors from each cell line, by subcutaneous injection into immunodeficient mice, as described in Methods section (Supplementary Table 1). Although a common protocol was suggested for tumor production, local circumstances mandated some modifications to this protocol in each case, particularly with respect to the particular strains of mice used as hosts. After cutting each tumor into several pieces, approximately half of them were randomly selected for histology, while the other half was processed to provide RNA for RNA-seq and TeratoScore analysis.

All PSC-derived tumors were classified as teratomas, since each contained tissues identified as derivatives of the three germ layers (Fig. 4a, b). Overall, a median of 10% (range, 5–30%) of the differentiated tissues observed were of endodermal derivation, 40% (range, 10–60%) represented tissues of mesodermal origin and 45% (range, 10–80%) represented tissues of ectodermal origin (Table 1 and Fig. 4c). Cells from all three embryonic germ

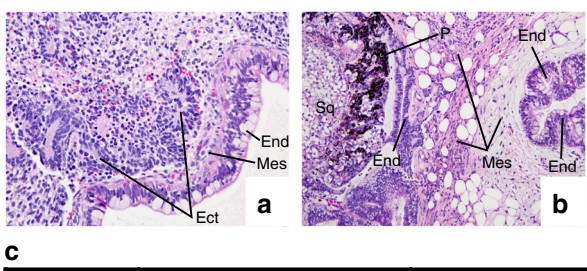

c

| Tissue | H9-ESC | | | | ESC | | | | iPSC | | | |
|---|---|---|---|---|---|---|---|---|---|---|---|---|
| | Lab1 | Lab2 | Lab3 | Lab4 | Lab1 | Lab2 | Lab3 | Lab4 | Lab1 | Lab2 | Lab3 | Lab4 |
| **Ectoderm** | | | | | | | | | | | | |
| Neural | 4/4 | 3/3 | 1/3 | 2/3 | 3/3 | 5/6 | 5/5 | 5/5 | 6/6 | 3/3 | 3/3 | 3/3 |
| pig. epith. | 4/4 | 3/3 | 1/3 | 2/3 | | 4/6 | 3/5 | 1/5 | 4/6 | 1/3 | 1/3 | 1/3 |
| sq. epith. | | 2/3 | 2/3 | | | | | | 4/6 | 2/3 | | |
| choroid pl. | 4/4 | | 3/3 | | | | | | | | | |
| **Mesoderm** | | | | | | | | | | | | |
| Cartilage | | 3/3 | 1/3 | 1/3 | 1/3 | 4/6 | 3/5 | 1/5 | 4/6 | 1/3 | 2/3 | 3/3 |
| Bone | | 1/3 | 1/3 | | | 3/6 | | | | | 2/3 | |
| Stroma | | | | 2/3 | | | | | | | | |
| Fat | | | 2/3 | | | | | | | | | |
| Mesenchy. | 4/4 | | 2/3 | 1/3 | | 3/6 | | 2/5 | 3/6 | 1/3 | | |
| Muscle | | | 2/3 | | | | | | | | | |
| **Endoderm** | | | | | | | | | | | | |
| Glands | 4/4 | | 1/3 | 2/3 | 3/3 | 2/6 | 2/5 | 3/5 | 2/6 | 1/3 | 1/3 | 1/3 |
| Ducts | | 3/3 | 2/3 | | 1/3 | 3/6 | | 3/5 | 3/6 | | | |
| Intestine | | | | 1/3 | | | 2/5 | 2/5 | | | | |

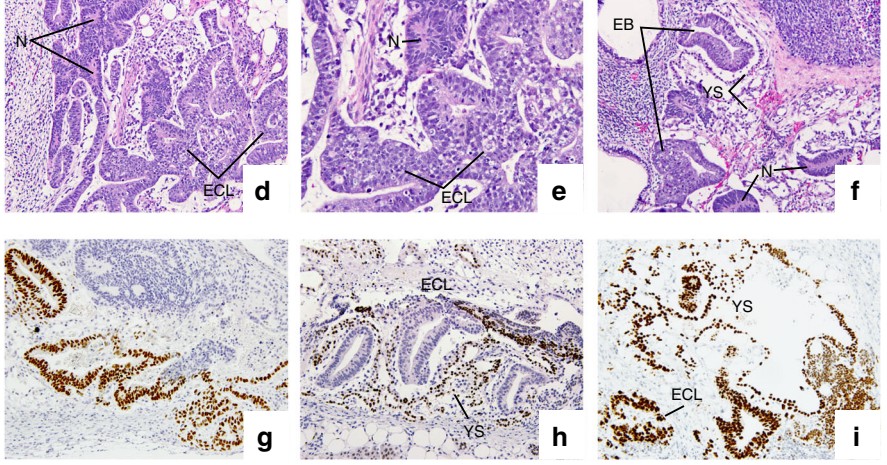

**Fig. 4** Histological evaluation of three embryonic germ layers and undifferentiated EC-Like and yolk sac elements in xenograft tumors. **a** Mucus secreting intestinal-like epithelium (End-endoderm), neural tube rosettes (Ect-ectoderm), and intervening stroma (Mes-mesoderm) (×240). **b** Intestinal-like epithelium (End-endoderm), surrounded by connective tissue, smooth muscle and fat cells (Mes-mesoderm). The left outer rim of mesoderm is lined by intestinal-like epithelium (End-endoderm). To the left there is pigmented epithelium (P), corresponding to retina (Ect-ectoderm), and a nest of glycogen rich squamous epidermal cells (Sq) (×120). **c** A summary of tissue types recorded per individual tumor piece surveyed from each laboratory; at least two pieces of each tumor were examined. **d** Lower magnification view of a teratoma containing undifferentiated stem cells (EC-Like, ECL), identified as embryonal carcinoma-like (ECL) cells, neural tube-like rosettes (N) and non-descript stromal cells (×120). **e** Higher magnification of the same xenograft. Undifferentiated ECL cells (ECL) are arranged into anastomosing cords. Dark dot-like cells are undergoing apoptosis. Compare the loosely structured chromatin of the ECL cells with the dark nuclei containing condensed chromatin in the neural rosettes (N) (×240). **f** Two embryoid bodies (EB) forming tubes lined by ECL cells, separated by a space from the surrounding yolk sac epithelium (YS). Both embryoid bodies contain prominent apoptotic bodies. Note the loosely textured yolk sac (YS) corresponding to the connective tissue that runs between the yolk sac and the blastocyst (magma reticulare) of early human embryos (×120). **g** Antibody to OCT3/4 staining ECL cell nuclei. **h** Antibody to the zinc-finger protein ZBTB16 reacts with the nuclei of yolk sac cells around three cylinders of ECL cells. **i** Antibody to SALL4 staining ECL cell nuclei and also the yolk sac (YS) cells in their vicinity

layers were found in the teratomas, derived from both ES and iPS cell lines produced by each of the laboratories. Although all teratomas contained derivatives of the three embryonic germ layers, in fact only a fairly narrow range of tissues was routinely identified. Neural tube-like structures, pigmented epithelium and squamous epithelium accounted for most ectoderm, cartilage, connective tissue, and bone for most mesoderm, and glandular, ductal and intestine tissue for most of the endoderm (Fig. 4c).

Some teratomas also contained areas of undifferentiated cells, which we designated as embryonal carcinoma-like (ECL) cells, some exhibited areas of yolk sac elements, and some contained cells in some areas organized into EB like structures (Fig. 4d–f). The histological identification of the ECL was confirmed by

immunostaining for expression of OCT3/4 (POU5F1) (Fig. 4g) and the yolk sac cells by immunostaining for ZBTB16 (Fig. 4h)[36]. As expected, SALL4 expression was found in both yolk sac and ECL cells[37,38] (Fig. 4i). The initiating PSCs in teratomas may differentiate into derivatives of mature elements of all three germ layers, and into extraembryonic elements, such as yolk sac[12], or the PSCs may proliferate in which case they may be noted as ECL cells, suggesting a potential malignant phenotype. In the clinical pathology of germ cell tumors (GCT), embryonal carcinoma and yolk sac elements are frequently found in malignant

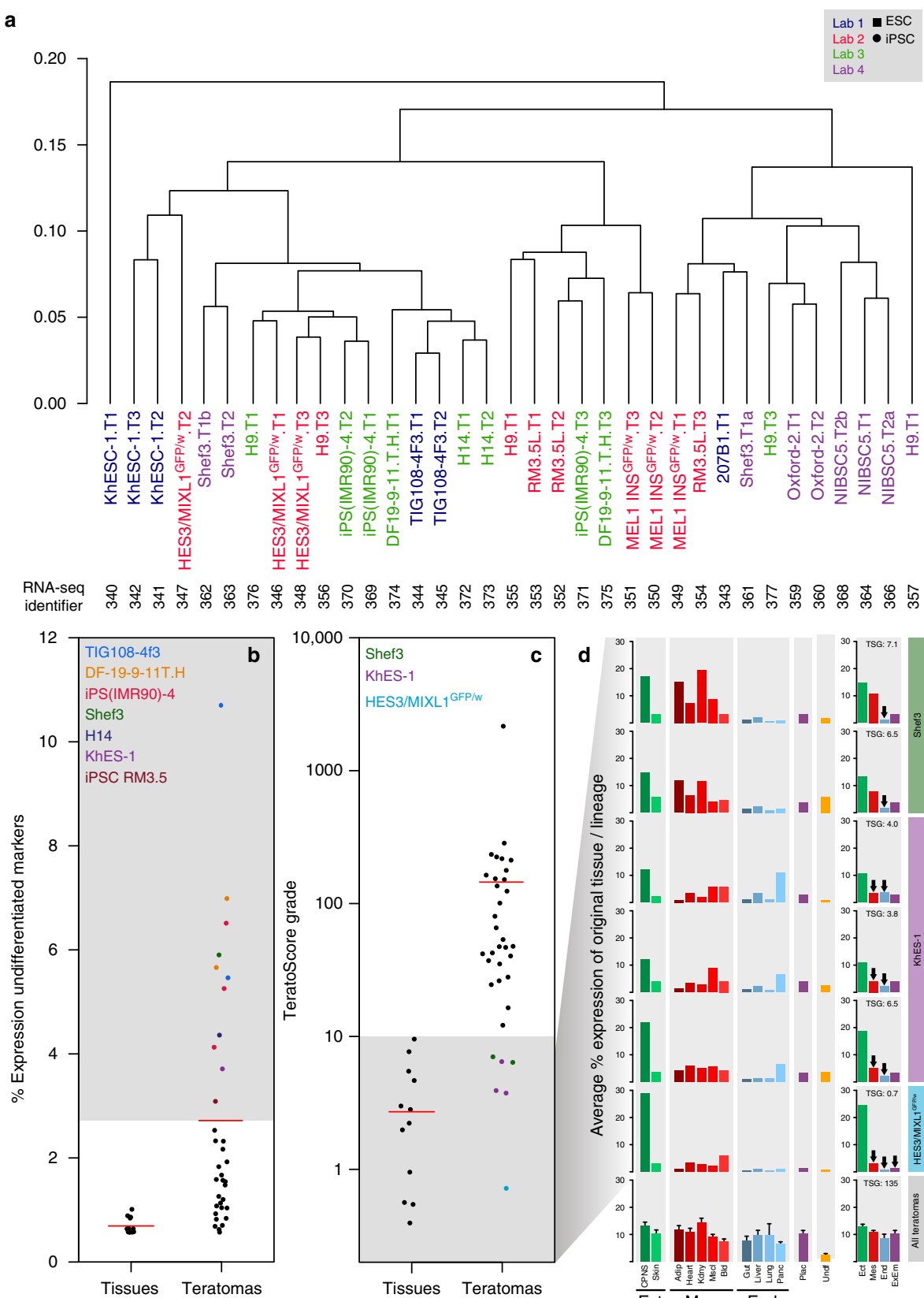

teratocarcinomas, (reviewed in ref. [4]), while yolk sac and immature neural elements are commonly associated with malignant transformation in teratomas of childhood[39–42]. It has been proposed that the experimental teratomas produced by both mouse and human ES and iPS cells are more akin to GCT of the newborn (type 1 GCT), than to those of the adult (type 2 GCT)[43]. This distinction correlates with the diploid or near diploid karyotypes of most ES and iPS cells, in contrast to the grossly aneuploid karyotypes of human EC cells from adult germ cell tumors. In the experimental teratomas at hand we find it noteworthy that even when these potentially malignant elements were observed, robust differentiation into tissues derivatives of all three germ layers was also seen within the same tumor. Histologic evidence alone does not permit a definitive conclusion as to whether the finding of ECL cells intermingled with yolk sac elements is indicative of the malignant potential of a subset of the PSCs tested, but it is certainly a cause for concern. Although, most teratoma histological sections include differentiated structures from the three embryonic germ layers, an elaborate analysis of cell type would require further experimental investigation.

RNA samples were extracted from 44 of the 58 tumors prepared. Of these, 35 samples passed the quality control tests for RNA quantity, purity and integrity required for RNA-seq and further analysis. These samples represented tumors derived from 12 different independent cell lines (6 ESC, 6 iPSC), as well as tumors derived from H9 in three of the four laboratories. An initial unbiased hierarchical clustering of all gene expression data was performed (Fig. 5a). Although tumors derived from the same cell line sometimes clustered together, sometimes they did not, and data from tumors of the different cell lines, be they ES- or iPSC-derived, as well as those from H9-derived tumors, even if from the same laboratory, were scattered throughout the dendrogram suggesting there was no obvious laboratory effect (Fig. 5a).

To assess whether there were residual undifferentiated PSC within the teratomas, we queried the RNA-seq datasets for expression of ten undifferentiated PSC markers (Supplementary Table 3). These marker genes were initially selected based on results previously published by the ISCI[44] and include several markers also expressed by yolk sac endoderm cells[45–47]. When expression of these genes in the teratomas was compared to that of cultured undifferentiated PSC, their mean expression was found to be 2.5% of that in the PSCs (Supplementary Table 3). Nevertheless, eleven teratoma samples, originating from seven different PSC lines (KhES-1, TIG108 4f3, RM3.5, H14, DF19-9-11T.H, IPS(IMR90)-4, Shef3), exhibited substantially higher average expression levels of these 10 markers, suggesting the presence of undifferentiated PSCs and/or yolk sac elements (Fig. 5b). Those teratomas showing elevated

expression of these marker genes also clustered in a principal component analysis (Supplementary Fig. 3). Of these lines, TIG108 4f3 had been previously classified as 'differentiation defective', and KhES-1 as 'intermediate defective', in an assay that assessed the persistence of undifferentiated, OCT3/4$^+$ (POU5F1) cells after a defined period of specific neural induction in vitro[48]. In concert with that report we noted that TIG108 4f3 tumors showed higher levels of the stem cell markers than KhES-1 tumors (Fig. 5b). Teratomas derived from five of these seven cell lines were found by histological analysis to contain ECL cells (KhES-1 and TIG108 4f3) and/or yolk sac cells (KhES-1, TIG108 4f3, H14, DF19-9-11T.H, Shef3) (Table 1). Overall, these results suggest that many of the teratomas contained differentiated derivatives of extra-embryonic membranes and potentially ECL cells.

The TeratoScore algorithm enables the use of teratoma gene expression to provide a quantitative analysis of the ability of a PSC line to differentiate[26]. This analysis quantifies tissue-specific-gene expression within heterogeneous PSC-derived teratomas, thus providing an estimation of teratoma tissue composition. Since TeratoScore was originally designed for microarray analysis, it was adapted in this study to analyze RNA-seq data. Similar to the original TeratoScore calculation, a 100-gene signature was created by identifying genes expressed in teratomas and specific to tissues representing derivatives of the three embryonic germ layers and the extra-embryonic membranes (Methods section; Supplementary Data 4). Comparing expression of these genes in a teratoma to their respective expression level in normal tissues provides an estimate for the existence of cells from each tissue within the tumor, as well as a lineage expression proportion. By calculating the expression values from the different lineages, the TeratoScore provides a unified grade that weighs the different tissue-specific expression within a teratoma and provides an estimate of the ability of a PSC line to differentiate (Methods section). As expected, each individual normal tissue yielded a high-expression level of its specific cell type and lineage (Supplementary Fig. 4), yet a low unified TeratoScore grade (Fig. 5c). In contrast, teratomas show a relatively high score for all cell types (Supplementary Fig. 4) and lineages, and also higher TeratoScore grades (Fig. 5c). A TeratoScore grade >10 was deemed sufficient to determine that a given tumor was initiated from a PSC line capable of differentiating toward derivatives of three germ layers in a relatively evenly distributed fashion, since no normal tissue exceeded this threshold (Fig. 5c). However, of the 35 teratomas tested, six samples originating from three PSC lines (Shef3, KhES-1, and HES3 *MIXL1*$^{GFP/w}$) did not reach this threshold (Fig. 5c). A closer look at the expression patterns from these teratoma samples revealed higher expression of neuroectodermal markers in KhES-1 and HES3 *MIXL1*$^{GFP/w}$ compared to

**Fig. 5** Teratoma RNA-seq expression data analysis. **a** Unsupervised hierarchical clustering analysis of RNA-seq expression of teratomas from four different laboratories (calculated using complete linkage and Spearman correlation distance). Tumors from the same laboratory appear in the same color. Label numbers (T1, T2, etc.) indicate teratoma replicates. Specific RNA-seq sample identifiers are indicated below the sample names. **b** Mean relative expression of human undifferentiated PSC/yolk sac markers within teratomas and normal tissues calculated with respect to their expression in PSCs. Eleven teratomas (highlighted by colored dots) showed an expression greater than teratoma overall average (2.5%). **c** TeratoScore grades, calculated from RNA-seq profiles of normal tissues and teratomas. Each grade represents expression of markers from the three embryonic germ layers and extra-embryonic membranes. Normal tissues provided a mean grade of 2.7 ± 0.2, while teratomas provided a mean grade of 145.0 ± 61.6. Six teratomas from three lines (Shef3, KhES-1 or HES3 *MIXL1*$^{GFP/w}$) provided a grade lower than 10, the threshold reflecting sufficient representation of all lineages. Samples with a low TeratoScore grade are highlighted. **d** Distribution of aberrant tissue expression in teratomas. Shef3- derived teratomas show a low expression of endodermal and placental markers, whereas KhES-1 and HES3 *MIXL1*$^{GFP/w}$ teratomas show high expression of ectodermal markers and low expression of all other lineage markers. Arrows designate lineages with distinctly low expression (<4% of mean expression ratio). (TSG: TeratoScore Grade; Ect: Ectoderm; Mes: Mesoderm; End: Endoderm; CPNS: Central and Peripheral Nervous System; Adip: Adipose Tissue; Kdny: Kidney; Mscl: Skeletal Muscle; Bld: Blood; Panc: Pancreas; Plac: Placenta; Undf: Undifferentiated Markers; ExEm: Extraembryonic). Error bars represent SEM

all other lineages and lower expression of endodermal markers in Shef3 compared with the other lineages (Fig. 5d).

Comparing these data with those from histologic analysis, we find that ectoderm-derived tissues were also found at moderately high levels in KhES-1 and HES3 *MIXL1*[GFP/w] teratomas (Table 1). The rather low levels of endoderm-derived tissues in Shef3-derived teratomas were also confirmed by histologic analysis. However, the high-ectoderm content for DF19- 9-11T.H-derived tumors seen in tissue sections was not flagged by the TeratoScore assay. Histological analysis has a long and well accepted history in anatomy and clinical practice, and is aided by the propensity of cells within teratomas to form organoid structures that may be more readily recognized than individual isolated cells. On the other hand, TeratoScore, based on analysis of RNA-seq data, has the potential to reveal the presence of a wider range of cell types, such as those that do not form readily identifiable structures. However, in the absence of an all-inclusive histological atlas of gene expression in all developmental stages from embryo to adult, TeratoScore could under- or over-estimate lineage composition depending on the particular tissues present. The micro-anatomical heterogeneity of teratomas presents a drawback for both approaches since the number of tissue sections that can practically be viewed is often limited; whereas, the sensitivity of RNA-seq suggests that cell types present in small proportions will be missed. Nevertheless, there was a good degree of concordance between the two approaches on samples from the same teratomas when identifying cells from the three embryonic germ layers.

On the other hand, there was not much agreement between the in vitro EB assays and the teratomas, assessed by both histology and TeratoScore, in uncovering any apparent lineage bias of individual PSC lines. RNA from the teratoma samples was also analyzed by qRT-PCR using the same gene panel as for the EBs. In general, there was little concordance between the expression patterns of these genes in the EBs and teratomas derived from the corresponding PSC lines; even at 16 days of EB differentiation the teratomas did seem to show a higher tendency toward endoderm differentiation than the corresponding EBs (Table 1, Fig. 3c). This appeared to be the result of high expression of particular individual marker genes (foremost *GCG* and *FABP2*; see Supplementary Fig. 2d) and may not actually correspond to the presence of differentiated endodermal tissues, which was low according to the histologic analysis (Table 1). There was also no concordance between the persistence of undifferentiated stem cells in the teratomas (Table 1) and their persistence in the EBs after 16 days of differentiation (Supplementary Fig. 5): for example, although EBs formed under neutral conditions from KhES-1, HES3 *MIXL1*[GFP/w], H9 (Lab 1) and TIG108 4f3 showed evidence of similar levels of persisting undifferentiated cells, ECL cells were only identified in KhES-1, TIG108 4f3, and 201B7 tumors. Moreover, in the same EB-formation conditions KhES-1, 201B7, RM3.5C, and H9 (from Labs 2 and 4) had reduced propensities to form mesoderm and endoderm, an apparent bias not recapitulated in the teratoma assay (Table 1). Indeed, there was a tendency for the teratoma assays to highlight even greater ectodermal and less endodermal differentiation than the EB assays. Differences in the in vitro versus the in vivo environment, and in the timeline of the assays, likely account for these discrepancies. That is, xenograft tumor formation takes place over a number of weeks, after potentially undergoing complex interactions, both within the tumor and between the tumor and host tissue, whereas EB assays are performed within days of their formation and therefore assess much earlier stages of differentiation. Nonetheless, these two analytical approaches provide complementary information of pluripotency and differentiation potential.

## Discussion

In this global collaboration, under the auspices of the ISCI, we have compared three types of assay that featured independent in vivo and in vitro analyses of samples prepared under standardized conditions in four highly experienced laboratories to assess the developmental potential of human PSCs. Each of these approaches does provide evidence of pluripotency, but each measures quite different endpoints, each with its own distinct limitations, and each provides markedly different insights into the behavior of the cells. PluriTest provides a good and facile screening tool to identify cell lines that deviate sharply from a pluripotent gene expression profile. The capacity for PluriTest to be readily revised, refined, and updated, as it was in this study, could be seen as an advantage, particularly as the technology for transcription profiling and bioinformatics analysis evolves. The assay was developed for predicting teratoma formation based on whole-genome analysis, but provides no direct information on potential differentiation biases, and has not been shown to identify cell lines that display signatures of malignancy. In vitro differentiation assays combined with bioinformatics scorecard analysis of genes representative of the three embryonic germ layers, provide a simple and direct biological readout. In contrast to the in vivo teratoma assays, such in vitro tests provide quantitative information on differentiation potential that can be readily assessed in an unbiased fashion and do not require a specialist for histologic interpretation. On the other hand, like PluriTest, these assays are currently unable to identify cell lines that show biological behavior similar to that of transformed cells.

Prior to the emergence of large scale efforts to derive human iPS cells the teratoma assay was regarded as the gold standard in the field. The assay provides unequivocal evidence of a stem cell's capacity to differentiate (the formation of a wide range of tissues is monitored directly, as is the capacity for tissues to undergo histotypic organization). Due to the length and cumbersome nature of the assay and its requirements for animal usage and expert pathological assessment there are real limitations of the teratoma assay as a routine screening tool, and in practice, in this study and that of Bouma et al.[49], teratoma formation did not yield any greater discrimination concerning the differentiation potential of PSC lines than the in vitro assays. However, the teratoma assay was the only one which provided evidence of malignant potential of some of the PSCs. This is an important parameter that impacts on both the experimental and clinical use of the cells. Though the presence of undifferentiated stem cells, yolk sac elements and primitive neuroectoderm are indicative of malignancy in clinical germ cell tumor histopathology, their biological significance has not been assessed in the context of PSC xenografts. Future studies could undertake the prospective isolation and re-transplantation of such cells from xenografts, with a view toward determining their potential for initiation of tumors with histologic features of malignancy, including invasion and metastasis. Furthermore, in the clinical setting of childhood germ cell tumors, which resemble those derived from PSC, malignant behavior can be attributed not only to undifferentiated PSC but also to differentiated elements including yolk sac and primitive neuroectoderm. Thus, while it is essential to eliminate undifferentiated PSC from products destined for clinical use, safety assessment must take into consideration the possibility of malignancy arising from such differentiated tissues.

It is also interesting to note in this context that in contrast to other studies in which correlations with karyotypic abnormalities have been suggested to influence malignant potential[28,50–53], no such correlation was evident in the current study. Indeed, the teratomas in which ECL cells and/or yolk sac were identified derived from PSC lines with apparent diploid karyotypes. On the other hand, PSC lines found to be mosaic for abnormalities, by

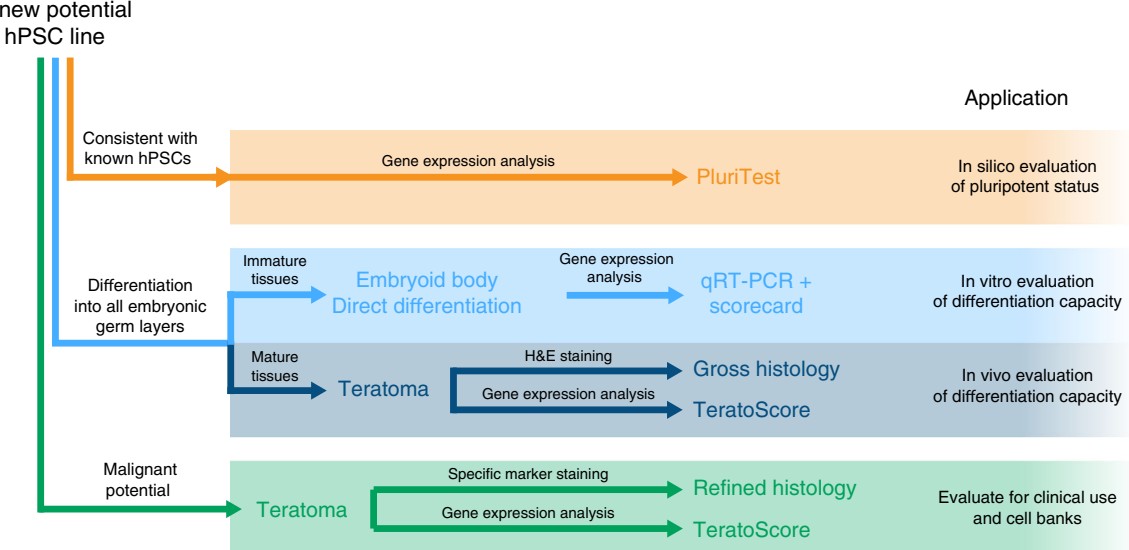

**Fig. 6** Proposed strategy to analyze new human pluripotent stem cell lines depends on the information required. To first determine whether or not a cell line is pluripotent (orange lines), its signature can be compared to that of known pluripotent cells' signatures using PluriTest. To confirm whether that cell line (blue lines), is capable of differentiating into derivatives of all three embryonic germ layers in vitro embryoid body (EB) formation in 'neutral' differentiation conditions, or by specific lineage-promoting differentiation conditions, combined with bioinformatic scorecard analysis, should be sufficient. If necessary differentiation to specific mature cells types may be also assessed in vivo by xenografting and teratoma formation followed by either histological analysis or RNA-seq analysis using Teratoscore. But to evaluate whether the cell line in question might have malignant potential (green lines) careful examination of histological sections of the teratoma using antibodies to specific markers or by focusing the RNA-seq and TeratoScore on specific markers associated with a malignant phenotype is suggested

karyotyping and/or e-Karyotyping, and teratomas shown to be abnormal by eSNP-karyotyping, did not exhibit ECL cells and robustly differentiated into derivatives of all three germ layers. A causative relationship between genetic changes in PSC lines and the presence of ECL, with or without yolk sac elements, in teratomas requires further investigation and the significance of these findings for future clinical applications remains to be established.

Based on our results, we suggest the choice of pluripotency assay depends upon the level of assessment required for a particular application (Fig. 6). Analysis of gene expression in the pluripotent state by PluriTest can be used as a screen to identify rapidly cells that also meet other criteria of pluripotency. PluriTest was designed to be continuously improved: as data from well-characterized training sets of cell lines that show defective differentiation or malignant behavior are added, PluriTest gains power to discriminate subtler characteristics of pluripotent cells. Meanwhile, if direct and quantitative confirmation of differentiation capacity is required, we recommend in vitro spontaneous and directed EB differentiation combined with bioinformatic scorecard analysis, which provides a rapid and facile alternative to the teratoma assay, and one that can be accepted as evidence of pluripotency for purposes of standard cell line characterization. Further, consideration of indicator gene panels taking into account key nodes in gene regulatory networks, may provide better identification of differentiation outliers and future assessment of the capacity for morphogenesis in 3D organoid type cultures in vitro might also prove helpful. At present, we suggest that, independent of other assays used to characterize these cells, it is prudent to carry out the teratoma assay on cells destined for clinical use. Cell banks should consider this option carefully as a part of their standard characterization protocol, particularly for widely used cell lines. The application of TeratoScore provides a more quantitative approach to the readout of the teratoma assay, compared to histologic analysis alone; however, we strongly recommend further research efforts to identify in vitro surrogate biomarkers indicative of malignant potential. Future comparison of results from teratoma assays with in vitro studies, and with genomic analyses of cell lines that yield teratomas with malignant elements, may provide simpler approaches, including in vitro surrogate genetic and epigenetic biomarkers, to identify cell lines with malignant potential.

## Methods

**Cell culture**. Each participating laboratory was asked to select three PSC lines (Supplementary Table 1) to analyze together with a PSC line, H9 (WA09)[12], that was used in common in all laboratories. The cells were grown according to the standard conditions typically used in each participating laboratory (Supplementary Table 1).

**e-Karyotyping**. Gene expression profiles of the undifferentiated samples were analyzed using Illumina HT12 microarray platform as described for PluriTest, below. Annotations of the microarray platform probes were obtained from the Illumina website (http://www.illumina.com/). Probe sets were organized by their chromosomal location, and their expression values were log2-transformed. Probe sets without annotated chromosomal locations were removed. An expression threshold was defined according to the levels of the upper third highest expressing probes, and probes with lower expression were elevated to this threshold. Probe sets not expressed in over 20% of the samples were removed to decrease expression noise. To obtain a comparative value, the median of each gene expression value across all samples was subtracted from the gene's expression value in each sample. This median also served as a baseline to examine expression bias. The 10% most variable genes, calculated by the sum of squares of relative expression value for each gene, were removed during the analysis. Data were processed using CGH-Explorer (http://www.softgenetics.com/CGHExplorer.html). A moving-average plot was generated using the moving average fit tool, with windows of 300 genes.

**eSNP-karyotyping**. eSNP-karyotyping was performed as previously described[33]. Briefly, RNA-seq reads were aligned to the genome (assembly version GRCh38) using Tophat2[54] and SNPs were called using GATK HaplotypeCaller[55]. Called SNPs were filtered by read number, with SNPs expressed in <20 transcripts discarded, and minor allele frequency and allelic ratio (major to minor) was calculated for the whole transcriptome. For visualization, moving medians of the major to minor ratios were plotted along the moving medians of the chromosomal positions using a window of 151 SNPs.

**PluriTest**. PluriTest analysis was performed as previously described[18]. We used R3.2.1 together with lumi 2.20.2 and the original PluriTest workspace. Due to an

overall shift in PluriTest results from experiments performed with newer versions of the Illumina microarray platform, we added a correction-vector to the matrices used in the computation of the Pluripotency and Novelty Scores. We used the H9 samples available from all laboratories to correct the data toward the reference H9 normalization target used in the original PluriTest algorithm (Supplementary Figure 1). The shift-vector is simply the difference between the row-wise means of the H9 samples in the current dataset and a H9 reference sample used as the normalization target in the original PluriTest implementation[18]. Since, the shift-vector is not restricted to positive data, we relax the non-negativity condition and estimate the matrix calculation by replacing the multiplicative update used in the original PluriTest workspace with a standard linear regression. The modified algorithm was tested on the original training dataset to guarantee consistent results (Supplementary Figure 1). The scripts required to run the analysis are provided via the group GitHub repositories and PluriTest's website (www.pluritest.org). The PluriTest workspace is available at https://github.com/pluritest/pluritestCompared.

**Production of size-controlled embryoid bodies**. EBs were produced as previously described[56]. Briefly, cells were trypsinized, counted and re-seeded in 96 well u-bottomed plates at a density of 3000 cells per well in APEL media (Stem Cell Technologies, Vancouver, CA) supplemented with factors for four differentiation conditions: neutral (without any growth factors), ectoderm (10 μM dorsomorphin, 10 μM SB431542 and 100 ng/ml basic-FGF), endoderm (100 ng/ml Activin-A, 1 ng/ml BMP4) and mesoderm (20 ng/ml Activin-A, 20 ng/ml BMP4) differentiation. All growth factors were added once at the onset of differentiation, the medium was not changed and the EBs were left in suspension for the course of the experiment. Biological replicates of each cell line were differentiated and harvested at three time points (4, 10, and 16 days) into RNAlater (Life Technologies, USA) and stored at −80 °C for future use.

**RT-PCR gene expression analysis**. Total RNA was extracted and purified using the PicoPure RNA Isolation Kit (Arcturus Bioscience) and QCed with a 2100 Bioanalyzer (Agilent Technologies). The high-capacity cDNA Archive Kit (Thermo Fisher Scientific) was used to generate cDNA representative of the polyadenylated transcriptome. Preamplification of cDNA was performed using the TaqMan Pre-Amp Master Mix (Thermo Fisher Scientific) following manufacturer's instructions with 10 cycles of amplification. Each of the two sets of 96 Delta Gene assays were pooled for priming of the preamplification reaction (i.e., two independent pre-amplification runs for each RNA sample). Delta gene assays were designed and provided by the manufacturer (Fluidigm) and are listed in Supplementary data 6, 7. Real-time PCR was performed using these Delta Gene assays (Fluidigm), the preamplified cDNAs, and 96.96 Dynamic Arrays (Fluidigm) run on a Biomark HD Real-time PCR System (Fluidigm) following the Fast Gene Expression Analysis Using EvaGreen protocol (User Guide PN 68000088 J1) provided by the manufacturer (Fluidigm). Cycle Threshold (Ct) values were calculated using the instrument's software (Application Version 4.1.2; Fluidigm).

**Production of teratomas**. Teratomas were generated in immunodeficient mice according to a common protocol but necessarily modified to accommodate local laboratory circumstances (Supplementary Table 1) and governed by local animal experimental rules. After a suitable growth period, the tumors were excised and divided into several pieces. To ensure representation across the tumor, a random selection of half of the pieces of each tumor was placed in RNAlater (Life Technologies, USA) and frozen at −80 °C prior to shipping for RNA-seq analysis. The remaining half of the pieces were fixed in 10% formal-saline prior to processing for histological analysis.

**Histological analysis**. At least two different teratoma pieces were sampled from each PSC line injected. Serial sections from 2 to 10 different pieces of each tumor were examined by two investigators who estimated the amount of differentiation into tissues derived from all three germ layers. The presence of yolk sac, embryoid bodies (EB) and undifferentiated cells, classified as embryonal carcinoma-like cells (ECL), were also noted.

**Immunohistochemical staining**. Sections (4 μm) from formalin-fixed, paraffin embedded samples were subjected to immunohistochemical detection of ZBTB16, SALL4, and OCT3/4 (POU5F1). Briefly, after deparaffinization and rehydration, tissue sections were treated using either citrate buffer (ZBTB16) or Borg Decloaker (SALL4, OCT3/4, Biocare Medical, Concord, CA) for 5 min in a pressure cooker for antigen retrieval. Hydrogen peroxide (3%) was then applied to the sections to quench endogenous peroxidase activity. Sections were then incubated with primary antibodies against ZBTB16 (PLZF clone D-9; 1:50 dilution; Santa Cruz Biotechnology, Santa Cruz, CA, USA), SALL4 (Clone 954–1054; 1:100 dilution; Biocare Medical, Concord, CA, USA) and OCT3/4 (Clone SEM; prediluted; Biocare Medical, Concord, CA, USA) for 45 min. After extensive rinsing, all sections were incubated with anti-mouse HRP-labeled polymer (EnVision ™+ System, Dako, Carpinteria, CA, USA) for 30 min. Finally, the staining was visualized by DAB+ (Dako, Carpinteria, CA, USA). Immunohistochemical staining was performed

using the IntelliPATH FLX Automated Stainer at room temperature. A light hematoxylin counterstain was performed, following which the slides were dehydrated, cleared, and mounted using permanent mounting media.

**RNA-seq analysis**. RNA was purified as described in RT-PCR analysis (below) and the same teratoma RNA samples were used in both the RT-PCR and RNA-seq experiments. RNA-seq libraries were prepared using the RNA sample preparation kit v2 (Illumina) according to the manufacturer's standard protocol. Briefly, polyadenylated RNA was first purified from total RNA was first purified through oligo-dT attached magnetic beads using two rounds of purification. Poly(A) RNA was subsequently fragmented and primed with random hexamers for cDNA synthesis. First strand cDNA synthesis was for 50 min at 42 °C using SuperScript II reverse transcriptase. After second strand cDNA synthesis, multiple indexing adapters were ligated, and libraries quality controlled with 2100 Bioanalyzer (Agilent Technologies), normalized and pooled prior to sequencing. Libraries were subjected to 101 base pair-end multiplex sequencing on an Illumina HiSeq 2000 in high-output mode. Samples were multiplexed (7–8 samples per lane) resulting in an average depth of 58 million reads per sample. Reads were aligned (human reference genome hg19) and transcripts counted using Tophat and Cufflinks. Data for analysis was expressed as FPKM values (Supplementary data 8, 9)

**TeratoScore analysis**. Since the original TeratoScore analysis[26] was performed on DNA microarray data, it was necessary to adapt the algorithm to the analysis of RNA-seq data. Briefly, a 100-gene scorecard of tissue-specific genes representing the three embryonic germ layers and extra-embryonic membranes was established (Supplementary data 4). By comparing RNA-seq expression data of 14 human body tissues, we identified genes with high tissue-specificity (expressing over 8-fold higher in a given tissue, compared to the mean of all other tissues). The expression of these genes was then compared between human PSCs and teratomas, validating their enrichment in differentiated cells (expressing over fourfold higher in teratomas compared to PSCs). Tissue specificity was finally validated with Amazonia! (http://amazonia.transcriptome.eu), with a requirement for distinct tissue expression (an order of magnitude over all, or most, other tissues)[57]. The RNA-seq data utilized in this analysis were obtained from the following sources: 13 human body tissues were obtained from The Genotype-Tissue Expression project (GTEx, http://www.gtexportal.org)[58] (Supplementary data 5). A minimum of five samples from each tissue was used to calculate a baseline expression, with samples chosen by the shortest ischemic time and highest RNA quality (Supplementary data 5). RNA-seq expression data for extra-embryonic tissues and human PSCs were obtained from the NIH Sequence Read Archive (SRA) (http://www.ncbi.nlm.nih.gov/sra) (Supplementary Data 5). Teratoma gene expression was obtained from 4 karyotically normal teratomas from the ISCI cohort (Supplementary data 5). Gene lists for central and peripheral nervous system, and for small intestine and colon were each merged together, as their specific-gene expression was similar. To generate the TeratoScore output, average expression for each lineage was calculated as the mean expression of all genes representing that lineage. TeratoScore grades were calculated as the multiplication of these means and dividing this product by 100.

**RT-PCR analysis of EB differentiation**. Data from all Fluidigm plates were collected (Supplementary Data 2) and analyzed in R (https://www.r-project.org). Low-quality datasets were removed (<33% of expected genes detected) and the raw Ct values were capped at 35, scaled to the control genes (ACTB, GAPDH), quantile-normalized[59,60]. To ease interpretation, we inverted the normalized numbers by subtracting them from the maximum (Ct = 35), resulting in numbers in which greater values represent stronger expression. For all further analysis and plots, we selected 15 marker genes per lineage based on effect size during differentiation in the EB assays (Supplementary Table 2). To this end, we calculated the rank of each gene in the comparison between expression measurements at day 16 (compared to day 0) per culture condition, taking the median across all cell lines. We then picked the 15 top-ranked genes for each condition as markers for the respective lineage, and the fifteen genes with the lowest average rank (i.e., those that were down-regulated, on average, as response to culture conditions) as markers for undifferentiated cells. Scorecard analysis was afterwards performed as previously described[22]. Briefly, we calculated a parametric gene set enrichment analysis on moderated t-scores for comparisons between each set of replicates (per cell line, time and condition) and all data at day 0. We then used a modified gene set enrichment analysis to examine the over representation of lineage markers in the gene lists ordered by these t-scores[61–63].

**Data availability**. The authors declare that all data supporting the findings of this study are available within the article and its supplementary information files or from the corresponding author upon reasonable request. Data from Illumina arrays (Pluritest), Fluidigm PCR and RNA-seq experiments have been deposited in the GEO database under accession code GSE97964.

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

## Acknowledgements

The International Stem Cell Initiative is funded by The International Stem Cell Forum (MRC grant reference MC_qA137863). A.G.E. and E.G.S. are Research Fellows of the NHMRC. Work in their laboratories was supported by grants from the Australian Stem Cell Centre, Stem Cells Australia, the Juvenile Diabetes Research Foundation and the National Health and Medical Research Council (NHMRC) of Australia. C.B. is supported by a New Frontiers Group award of the Austrian Academy of Sciences and by an ERC Starting Grant (European Union's Horizon 2020 research and innovation programme, grant agreement n° 679146). F.H. is supported by a postdoctoral fellowship of the German Research Council (DFG, HA 7723/1-1). F.-J.M. and B.M.S. were supported by grants from the BMBF (13GW0128A and 01GM1513D), from the Deutsche Forschungsgemeinschaft: German Research Foundation; DFG MU 3231/3-1 and from the DFG within the framework of the Schleswig-Holstein Cluster of Excellence, EXC 306 Inflammation at Interfaces. J.F.L. is supported by the California Institute for Regenerative Medicine grants RT3- 0765, GCIR-06673-A, and DISC2-09073 and NIH R01 NS092042-02. LH, OO and GS, comprising the UK Stem Cell Bank team, were supported by the Phase IV UK Stem Cell Bank grant from the Medical Research Council and the Biotechnology and Biological Sciences Research Council grant reference MR/L01324X/1. M.F.P. was supported by the Australian Research Council SRI10001002. O.B. was supported by the German Federal Ministry of Education and Research (01EK1603A and 01ZX1314A) and the European Union's Horizon 2020 Research and Innovation Programme (667301). P.W.A. and C.L.M. also received funding from the European Community's Seventh Framework Programme (FP7/2007–2013), under grant agreement no 602423, and Horizon 2020 Research and Innovation Programme, under grant agreement no 668724. P.W.A. and I.B. were members of the Pluripotent Stem Cell Platform, a consortium funded by a grant from the UK Regenerative Medicine, MRC Reference MR/L012537/1. I.B. is supported by the Medical Research Council, MRC Reference MR/N009371/1. Y.A. is a Clore Fellow and N.B. is the Herbert Cohn Chair in Cancer Research. This research is part of Y.A. PhD thesis and was partially supported by The Rosetrees Trust and by The Azrieli Foundation. We thank Fluidigm for their reagent contributions towards the RT-PCR gene expression studies.

## Author contributions

P.W.A. co-ordinated the project, and together with N.B., O.B., B.B.K., C.L.M., S.K.W.O., M.F.P., B.R., J.R. and G.N.S. formed the Steering Committee that provided overall scientific management. N.N., H.S., K.T. and S.Y. provided material from the culture and differentiation of hPSC at Kyoto University. A.E., R.M., S.M., E.S.Ng., K.S. and Ed.S. provided material from the culture and differentiation of hPSC at Murdoch Childrens Research Institute and Monash University. L.E.H., O.O'S. and G.N.S. provided material from the culture and differentiation of hPSC at the UK Stem Cell Bank. T.F.A. and I.B. provided material from the culture and differentiation of hPSC at the University of Sheffield. J.B., D.F. and T.E.L. provided material from the culture and differentiation of hPSC at WiCell. I.D., B.B.K. and D.S. carried out Histology. T.X.H., J.L., J.S.M. and P.R. carried out RNA-seq, qRT-PCR, microarray analyses. Y.A. and N.B. carried out TeratoScore analysis and e-karyotyping and eSNP-karyotyping. J.F.L., F.-J.M. and B.S. carried out PluriTest analysis. F.H. and C.B. carried out analysis of E.B. differentiation. P.J.G. curated the data. P.W.A., Y.A., P.J.G., F.H., B.B.K. and M.F.P. drafted the manuscript.

## Additional information

**Competing interests:** C.L.M. is a co-founder of Pluriomics bv (from Sept 2017 Ncardia). N.N. is a director and shareholder of hPSC-related companies Stem Cell and Device Laboratory, Inc. and Kyoto Stem Cell Innovation, Inc. He is also a shareholder of ReproCELL, Inc. O.B. is a co-founder and owns equity of LIFE and BRAIN GmbH. S.Y. is a scientific advisor of iPS Academia Japan without salary. K.T. is a member of the scientific advisory board of I Peace Inc. without salary. The remaining authors declare no competing interests.

## The International Stem Cell Initiative

Thomas F. Allison[1,2], Peter W. Andrews[1], Yishai Avior[3], Ivana Barbaric[1], Nissim Benvenisty[3], Christoph Bock[4,5,6], Jennifer Brehm[7], Oliver Brüstle[8], Ivan Damjanov[9], Andrew Elefanty[10,11], Daniel Felkner[7], Paul J. Gokhale[1], Florian Halbritter[4], Lyn E. Healy[12,13], Tim X. Hu[14], Barbara B. Knowles[15,16], Jeanne F. Loring[17], Tenneille E. Ludwig[7], Robyn Mayberry[10,11], Suzanne Micallef[10,11], Jameelah S. Mohamed[14], Franz-Josef Müller[18,19], Christine L. Mummery[20], Norio Nakatsuji[21], Elizabeth S. Ng[10,11], Steve K.W. Oh[22], Orla O'Shea[12], Martin F. Pera[23,24,25,26,27], Benjamin Reubinoff[28], Paul Robson[14,29,30], Janet Rossant[31,32], Bernhard M. Schuldt[19], Davor Solter[16,33], Koula Sourris[10,11], Glyn Stacey[12,34], Edouard G. Stanley[10,11], Hirofumi Suemori[35], Kazutoshi Takahashi[36] & Shinya Yamanaka[36,37]

[1]Centre for Stem Cell Biology, Department of Biomedical Science, University of Sheffield, Western Bank, Sheffield S10 2TN, UK. [2]UCLA School of Medicine, Department of Biological Chemistry, 615 Charles E. Young Drive South, Los Angeles, CA 90095, USA. [3]The Azrieli Center for Stem Cells and Genetic Research, Department of Genetics, Silberman Institute of Life Sciences, The Hebrew University, Jerusalem 91904, Israel. [4]CeMM Research Center for Molecular Medicine of the Austrian Academy of Sciences, 1090 Vienna, Austria. [5]Department of Laboratory Medicine, Medical

University of Vienna, 1090 Vienna, Austria. [6]Max Planck Institute for Informatics, Saarland Informatics Campus, 66123 Saarbrücken, Germany. [7]WiCell Research Institute Inc., 504S. Rosa Rd., Suite 101, Madison, WI 53719, USA. [8]Institute of Reconstructive Neurobiology, LIFE and BRAIN Center, University of Bonn Medical Faculty, Sigmund-Freud- Strasse 25, 53127 Bonn, Germany. [9]Department of Pathology, The University of Kansas School of Medicine, Kansas City, KS 66160, USA. [10]Murdoch Childrens Research Institute, Royal Children's Hospital, Flemington Road, Parkville, VIC 3052, Australia. [11]Department of Anatomy and Developmental Biology, Monash University, Clayton 3800, Australia. [12]UK Stem Cell Bank, Advanced Therapies Division, NIBSC-MHRA, Blanche Ln, South Mimms, Potters Bar EN6 3QG, UK. [13]The Francis Crick Institute, 1 Midland Road, London NW1 1AT, UK. [14]Genome Institute of Singapore, Singapore 138672, Singapore. [15]Jackson Laboratory, 600 Main Street, Bar Harbor, ME 04609, USA. [16]Siriraj Center of Excellence in Stem Cell Research, Mahidol University, Bangkok 10700, Thailand. [17]The Scripps Research Institute, 10550 North Torrey Pines Road - SP30-3021, La Jolla, CA 92037-1000, USA. [18]Department of Genome Regulation, Max-Planck-Institute for Molecular Genetics, Berlin 14195, Germany. [19]Zentrum für Integrative Psychiatrie, Universitätsklinikum Schleswig-Holstein, Kiel 24105, Germany. [20]Leiden University Medical Center, P.O. Box 9600, 2300 RC Leiden, The Netherlands. [21]Institute for Integrated Cell-Material Sciences, Kyoto University, Kyoto 606-8501, Japan. [22]Bioprocessing Technology Institute, 20 Biopolis Way, #06-01 Centros, Singapore, Singapore 138668. [23]Stem Cells Australia, The University of Melbourne, Parkville, VIC 3010, Australia. [24]Department of Anatomy and Neurosciences, The University of Melbourne, Parkville, VIC 3010, Australia. [25]Florey Neuroscience and Mental Health Institute, Parkville, VIC 3052, Australia. [26]Walter and Eliza Hall Institute of Medical Research, 1G Royal Parade, Parkville, VIC 3052, Australia. [27]The Jackson Laboratory, 600 Main Street, Bar Harbor, ME 04609, USA. [28]Department of Obstetrics and Gynecology, and the Hadassah Human Embryonic Stem Cells Research Center, The Goldyne-Savad Institute of Gene Therapy, Hadassah-Hebrew University Hospital, Jerusalem 91120, Israel. [29]The Jackson Laboratory for Genomic Medicine, Farmington, CT 06032, USA. [30]Department of Genetics and Genome Sciences, University of Connecticut Health Center, Farmington, CT 06032, USA. [31]Program in Developmental and Stem Cell Biology, Hospital for Sick Children, Toronto M5G 1X8, Canada. [32]Department of Molecular Genetics, University of Toronto, Toronto M5S 1A8, Canada. [33]Max Planck Society, Munich 80084, Germany. [34]The International Stem Cell Banking Initiative, 2 High Street, Barley, Herts SG8 8HZ, UK. [35]Laboratory of Embryonic Stem Cell research, Institute for Frontier Life and Medical Sciences, Kyoto University, 53 Kawahara-cho, ShogoinSakyo-kuKyoto 606-8507, Japan. [36]Center for iPS cell Research and Application, Kyoto University, Kyoto 606-8507, Japan. [37]Gladstone Institute of Cardiovascular Disease, San Francisco, CA 94158, USA

