## [Peer Review File · Nature Communications]

Reviewers' Comments:

Reviewer #1:

Remarks to the Author:

The manuscript compares different approaches for characterising the differentiation capacity of human pluripotent stem cells (hPSCs). For that, 4 groups located in different institutions have analysed the differentiation capacity of 13 hPSCs lines using 3 methods including Tertomas, EB/Scorecard and Pluritest. Each method confirmed the pluripotent state of the hPSC line used in this study. However, some method seems to be more useful than other for measuring specific characteristic. For example, teratomas could indicate potential tumorigenic potential.

Several aspects of the manuscripts have been reinforced. The data are very interesting while this coordinated work between leading groups in the stem cell field is impressive. The conclusions of their results are clearly explained as their importance. The figure 6 is very useful to get some directions for lab deriving new hPSCs or for group developing regenerative medicine applications. The manuscript will be very useful for the field and will definitely help a diversity of labs to choose the right test for characterising their hPSCs.

Reviewer #3:

Remarks to the Author:

In this study, the International Stem Cell Initiative (ISCI) compares three common approaches to assessing the developmental potential of human pluripotent stem cells (PSC). A systematic comparison of the methods for evaluating PSCs' pluripotency and differentiation potential is useful to guide future applications of PSCs. Most results reported in this study are not unexpected. However, it is nice to have a multi-lab assessment of the commonly used methods. It is also nice that the authors provide specific guidelines for choosing different assays based on their assessment. As such, the results reported in this study are valuable for the stem cell community.

The authors did a good job to address my previous questions and questions raised by other reviewers. I do not have other major concerns. There are a couple of minor issues the author should address.

1. Figure 3 Panel c legend: "The lineage scores for ectoderm (left), mesoderm (center) and endoderm (right) marker expression are shown separately". Here "ectoderm (right)" should be "endoderm (right)".

2. Figure 1b and Supplementary Figure S4: the texts in the plots are too small. A larger font size should be used.

Reviewer #5:

Remarks to the Author:

In the opinion of this referee, the authors have adequately addressed the concerns and this interesting paper now deserves to be accepted for publication.

REVIEWERS' COMMENTS:

Reviewer #1 (Remarks to the Author):

The manuscript compares different approaches for characterising the differentiation capacity of human pluripotent stem cells (hPSCs). For that, 4 groups located in different institutions have analysed the differentiation capacity of 13 hPSCs lines using 3 methods including Tertomas, EB/Scorecard and Pluritest. Each method confirmed the pluripotent state of the hPSC line used in this study. However, some method seems to be more useful than other for measuring specific characteristic. For example, teratomas could indicate potential tumorigenic potential.

Several aspects of the manuscripts have been reinforced. The data are very interesting while this coordinated work between leading groups in the stem cell field is impressive. The conclusions of their results are clearly explained as their importance. The figure 6 is very useful to get some directions for lab deriving new hPSCs or for group developing regenerative medicine applications. The manuscript will be very useful for the field and will definitely help a diversity of labs to choose the right test for characterising their hPSCs.

Reviewer #3 (Remarks to the Author):

In this study, the International Stem Cell Initiative (ISCI) compares three common approaches to assessing the developmental potential of human pluripotent stem cells (PSC). A systematic comparison of the methods for evaluating PSCs' pluripotency and differentiation potential is useful to guide future applications of PSCs. Most results reported in this study are not unexpected. However, it is nice to have a multi-lab assessment of the commonly used methods. It is also nice that the authors provide specific guidelines for choosing different assays based on their assessment. As such, the results reported in this study are valuable for the stem cell community.

The authors did a good job to address my previous questions and questions raised by other reviewers. I do not have other major concerns. There are a couple of minor issues the author should address.

1. Figure 3 Panel c legend: "The lineage scores for ectoderm (left), mesoderm (center) and endoderm (right) marker expression are shown separately". Here "ectoderm (right)" should be "endoderm (right)".

We have altered the legend accordingly.

2. Figure 1b and Supplementary Figure S4: the texts in the plots are too small. A larger font size should be used.

We have altered the figures accordingly.

Reviewer #5 (Remarks to the Author):

In the opinion of this referee, the authors have adequately addressed the concerns and this interesting paper now deserves to be accepted for publication.